# Glycemic Control and Metabolic Adaptation in Response to High-Fat versus High-Carbohydrate Diets—Data from a Randomized Cross-Over Study in Healthy Subjects

**DOI:** 10.3390/nu13103322

**Published:** 2021-09-23

**Authors:** Ville Wallenius, Erik Elebring, Anna Casselbrant, Anna Laurenius, Carel W. le Roux, Neil G. Docherty, Christina Biörserud, Niclas Björnfot, My Engström, Hanns-Ulrich Marschall, Lars Fändriks

**Affiliations:** 1Institute of Clinical Sciences, Department Surgery, Sahlgrenska Academy, University of Gothenburg, SE-413 45 Gothenburg, Sweden; erik.elebring@gu.se (E.E.); anna.casselbrant@gastro.gu.se (A.C.); anna.laurenius@vgregion.se (A.L.); christina.biorserud@vgregion.se (C.B.); niclas.bjornfot@vgregion.se (N.B.); my.engstrom@gu.se (M.E.); lars.fandriks@gastro.gu.se (L.F.); 2Metabolic Medicine, School of Medicine, Conway Institute, University College Dublin, Dublin 4, Ireland; carel.leroux@ucd.ie (C.W.l.R.); neil.docherty@ucd.ie (N.G.D.); 3Institute of Medicine, Department Molecular and Clinical Medicine, Sahlgrenska Academy, University of Gothenburg, SE-413 45 Gothenburg, Sweden; hanns-ulrich.marschall@gu.se

**Keywords:** glucose, insulin, metabolism, glucagon-like peptide-1, high-fat, high-carbohydrate, diet

## Abstract

Granular study of metabolic responses to alterations in the ratio of dietary macro-nutrients can enhance our understanding of how dietary modifications influence patients with impaired glycemic control. In order to study the effect of diets enriched in fat or carbohydrates, fifteen healthy, normal-weight volunteers received, in a cross-over design, and in a randomized unblinded order, two weeks of an iso-caloric high-fat diet (HFD: 60E% from fat) and a high-carbohydrate diet (HCD: 60E% from carbohydrates). A mixed meal test (MMT) was performed at the end of each dietary period to examine glucose clearance kinetics and insulin and incretin hormone levels, as well as plasma metabolomic profiles. The MMT induced almost identical glycemia and insulinemia following the HFD or HCD. GLP-1 levels were higher after the HFD vs. HCD, whereas GIP did not differ. The HFD, compared to the HCD, increased the levels of several metabolomic markers of risk for the development of insulin resistance, e.g., branched-chain amino acid (valine and leucine), creatine and α-hydroxybutyric acid levels. In normal-weight, healthy volunteers, two weeks of the HFD vs. HCD showed similar profiles of meal-induced glycemia and insulinemia. Despite this, the HFD showed a metabolomic pattern implying a risk for a metabolic shift towards impaired insulin sensitivity in the long run.

## 1. Introduction

The mechanisms underlying the increasing prevalence of obesity and type 2 diabetes (T2D) are poorly understood. In particular, there is an unresolved debate on the role of the interaction between the diet and the gut for the development of T2D [1,2,3]. It is known that the relative exclusion of fat or carbohydrates from food may influence glycemic regulation [1,2,3]. In order to be absorbed, carbohydrates are chemically degraded into smaller mono- or disaccharides before being absorbed in the upper part of the small intestine (mainly the duodenum and jejunum) [4]. Triglycerides are degraded to free fatty acids and glycerol before being absorbed. On their way through the intestinal epithelium, carbohydrates and fatty acids stimulate enteroendocrine cells to release gut peptides, e.g., glucagon-like peptide (GLP)-1 and gastric inhibitory peptide (GIP), that enhance insulin release from the pancreas and thereby prepare whole-body metabolism for the incoming nutrient load.

The handling of the ingested nutrient load offsets the resting nutritional balance and causes fluctuations in the levels of intermediary metabolites in plasma. It has been hypothesized that some intermediary metabolites may influence long-term metabolic outcomes, e.g., by having toxic effects on the β-cells of the pancreas [5]. In line with this, increased levels of some of these intermediary metabolites, e.g., branched-chain amino acids, have been shown to be strong predictors of the future risk of the development of metabolic disorders [6,7,8,9,10].

To understand these mechanisms in more detail, we investigated the effects of 2-week fat-dominated and carbohydrate-dominated diets in a cross-over study in healthy, normal-weight volunteers. The participants started with one of the diets according to randomization and, after a 2-week wash-out, continued with the other diet. At the end of each 2-week treatment, the subjects performed a mixed meal test (MMT). The primary endpoint was to compare MMT-induced glycemia and insulinemia after the HFD and HCD. The secondary endpoint was to study the effect of the diets on MMT-induced release of insulin, GLP-1 and GIP, as well as the levels of metabolomic markers that may predict the long-term risk of developing dysglycemia.

## 2. Materials and Methods

### 2.1. Study Setting, Participants and Study Design

This study was conducted at the Department of Surgery (Gastrosurgical Laboratory) at Sahlgrenska University Hospital in Gothenburg, Sweden, between February and December (winter–spring–summer–autumn–winter) 2014 (first screening to last visit).

Eligible normal-weight and healthy participants were screened by the study nurse and study physician, using a structured interview (focusing on general health, potential gastrointestinal disorders and previous abdominal surgery) as well as routine blood biochemistry. Inclusion criteria were: voluntary participation, self-reported good general health status, age between 18 and 65 years and body mass index (BMI) between 18 and 25 kg/m^2^. Exclusion criteria were: overweight or obesity (BMI > 25 kg/m^2^); history of drug abuse or smoking; use of prescription medications within the previous 14 days (with the exception of contraceptives); pregnancy or breast feeding, or potentially childbearing women not using adequate birth control; in the investigator’s judgment, clinically significant abnormalities at the screening examination or in the laboratory test results. The study was designed as a single-center, unblinded cross-over study in healthy volunteers obtaining controlled equicaloric diets either rich in carbohydrates (60% energy; E%) or rich in fat (60E%), administered over two 2-week periods in random order with an intervening wash-out period of a minimum of 2 weeks. The inclusion of study participants is depicted in the CONSORT diagram in Appendix A. The primary outcome measure in ClinicalTrials.gov was defined as mucosal surface enlargement factor, and the secondary outcome measures were defined as epithelial electrical current and mucosal electrical resistance in vitro, as well as glycemic control following the mixed meal test (MMT). The sample size was estimated based on previous data on epithelial surface enlargement [4]. A thirty percent difference was set as the criterion for biological relevance, and with an assumed standard deviation of 30%, alpha of 0.05 and 90% power, the calculated sample size was determined to be 12. As endoscopy is considered quite uncomfortable, reducing the willingness for a repeated procedure, allocation was increased to *n* = 17 to compensate for dropouts. In the current report, glycemic control and related data will be presented, and the other outcome measures will be presented in separate articles.

### 2.2. Enrolment and Randomization

Sixteen eligible subjects were enrolled and completed a baseline visit with assessment of body weight and height and fasting blood sampling for basic biochemistry, including plasma glucose and insulin concentrations, as illustrated in the CONSORT diagram in Appendix A. The participants were allocated to the two individually adapted (see below under Diets and Study Instructions) iso-caloric diets separated with a “wash-out” period of at least two weeks during which time participants were free to consume their habitual diet. The order of the study diets was randomized in blocks due to cooking logistics, and the arm starting with the high-carbohydrate diet became somewhat larger (*n* = 9 vs. *n* = 7). Randomization was performed by the study nurse who randomly assigned the patients to start the study in one of the study diet blocks.

### 2.3. Diets and Study Instructions

A registered dietician calculated the daily energy need of each participant based on height, weight, age, gender and physical activity using the Mifflin St Jeor equation [11]. Two iso-caloric diets; “high-fat diet” (HFD) and “high-carb diet” (HCD), with 60% of the energy contents from either fat or carbohydrates, respectively, were composed corresponding to the estimated energy need of each participant (three different energy levels were composed for each diet). The composition of each macro-nutrient was chosen so it corresponded to ordinary Swedish food standards and is specified in Appendix A.

Examples of daily menus of the HCD and HFD are provided in Appendix A. All meals, beverages and snacks needed for a 2-week period, some of them fresh and some deep frozen (but easy to heat in a microwave oven), were delivered on a weekly basis from the laboratory kitchen and hospital restaurant. The participants were instructed not to eat anything but the food and drinks provided by the laboratory but were free to ingest extra tap water if needed. To increase compliance, the participants filled in a food diary using a smartphone application and reported food intake, appetite perception and potential adverse events as well as potential ingestion of extra tap water, on a daily basis. The study participants were also instructed to refrain from alcoholic beverages and continue ordinary daily life activities including physical activity.

### 2.4. Mixed Meal Test

At the end of each 2-week diet period (day 12), the participants visited the laboratory after an overnight fast for a mixed meal test (MMT). A fasting blood sample was drawn via an intravenous forearm canula. The study participants were served a 600 kcal brunch (15E% protein; 31E% carbohydrates; 54E% fat, see Appendix A for details) followed by blood sampling at 15 (=immediately following meal intake), 30, 45, 60, 90 and 120 min after meal start. Blood samples were prepared for plasma and serum, aliquoted and frozen until analysis at −80 °C. The participants filled in a three-part (craving, hunger, satiation) visual analogue scale at premeal and 30 min and 120 min after the meal (Table 1).

HCD (high-carbohydrate diet), 2-week diet with 60E% from carbohydrates; HFD (high-fat diet), 2-week diet with 60E% from fat. Visual analogue scale with 0 = lowest perceived sensation, and 100 = highest perceived sensation. Values are mean (95% CI), *n* = 15, Wilcoxon’s test.

### 2.5. Biochemistry

All blood samples for biochemistry profiling were analyzed by the central laboratory of the Sahlgrenska University Hospital (accredited according to European norm en45001), with the exception of plasma glucose, serum insulin, serum GLP-1 and serum GIP.

### 2.6. Glucose, Insulin, GLP-1 and GIP Quantifications

Blood glucose was measured using StatStrips according to the manufacturer’s instructions (Nova Biomedical, Waltham, MA). Fasting serum insulin was measured by an electrochemiluminescence immunoassay, “eclia”, using a Cobas immunoassay analyzer according to the manufacturer’s instructions (Roche Diagnostics, Rotkreutz, Switzerland). Insulin, GLP-1 and GIP in serum samples taken after various times during the MMTs (0, 15, 30, 45, 60, 90 and 120 min for insulin, and 0, 30, 60 and 120 min for GLP-1 and GIP) were quantified using ELISA techniques according to the manufacturer’s instructions. Kits used were Human Insulin ELISA, Multi Species GLP-1 Total ELISA and Human GIP (total) ELISA (Merck Millipore, Darmstadt, Germany). HOMA-IR was calculated using the formula [fasting glucose (mmol/L) × fasting insulin (mU/L)/22.5].

### 2.7. Metabolomics Sample Preparation and NMR Data Acquisition and Processing

For metabolomics, the established SOPs of Bruker In Vitro for Diagnostic research (IVDr; Bruker, Billerica, MA, USA; https://www.bruker.com/products/mr/nmr-preclinical-screening.html (accessed on 18 October 2019)) were adhered to for sample preparation, NMR spectrometer QA and data acquisition. Serum samples retrieved during the MMTs were thawed for 30 min at room temperature before centrifugation at 2250× *g* at 4 °C for 2 min. An amount of 310 µL of buffer (75 mM NaH_2_PO_4_, 0.08% 3-(trimethylsilyl)propionic-2,2,3,3-d4 acid, 0.1% NaN_3_, pH 7.4 in 20% D_2_O) was distributed in each well of 96-well deepwell plates (Porvair, Sint-Job-in-’t-Goor, Belgium; cat no 219030) with an electronic stepper pipette (E3; Eppendorf, Hamburg, Germany). An amount of 310 µL of each sample was transferred to the deepwell plates with a SamplePro L Tube liquid handling robot (Bruker BioSpin, Billerica, MA, USA). The plates were shaken at 9 °C, 800 rpm, for 2 min before a brief spin down at 2250× *g*, 4 °C, for 1 min. An amount of 600 µL of each mixed sample was transferred to 5 mm SampleJet NMR tubes with the SamplePro L Tube robot. Samples were kept cold throughout the preparation. ^1^H NMR data were acquired on a 600 MHz Bruker Avance III spectrometer equipped with a 5 mm room temperature BBI probe and a cooled SampleJet sample changer. Briefly, 1D NOESY, CPMG and 2D J-resolved spectra were acquired at 37 °C with the standard pulse sequences noesygppr1d, cpmgpr1d and jresgpprqf, respectively, as defined in the IVDr SOP. Data were processed within TopSpin3.5pl7 (Bruker BioSpin, Billerica, MA, USA), and lipoprotein profiling was achieved through the B.I.LISA v1.0.0 method supplied by Bruker. R was used to peak pick features in NOESY and CPMG spectra with the speaq2 package [12], and multivariate analysis was performed within R on both metabolite and lipoprotein profiling data using a range of different packages including ropls [13] and MUVR [14].

### 2.8. Statistics

Non-parametric statistics were used as a rule because of the relatively small sample size. Paired analysis was used, and the study subjects acted as their own controls. For time profiles during the MMTs, repeated measures ANOVA with Sidak’s multiple comparison test was used. For statistical comparisons of AUCs during the MMTs, the Wilcoxon test was used. For correlation analyses, the Spearman non-parametric method was used, and correlation was tested between the calculated HOMA-IR (homeostatic model assessment—insulin resistance) at fasting and by using the AUC (area under the curve) for GLP-1, GIP or insulin, and for respective metabolites during the MMTs. A *p*-value of ≤0.05 was considered significant. All analyses were performed using Prism 8 for Mac OS X (GraphPad Software Inc. San Diego, CA, USA).

## 3. Results

### 3.1. The Study Group and General Responses

All demographic data and blood biochemistry variables were within normal ranges in all sixteen study subjects. One female participant had to be excluded after the first dietary period due to an unplanned pregnancy. Therefore, fifteen participants fulfilled the study protocol and were included in the analysis (Table 2, Appendix A). During the two 14-day study periods, the subjects consumed two iso-caloric diets, administered in a randomized order: a high-fat diet (HFD) or a high-carbohydrate diet (HCD), each with 60% of the energy contents from fat or carbohydrates, respectively. There was at least a fortnight “wash-out period” between the two study periods during which the subjects consumed ad libitum diets. The two study diets were generally well accepted and tolerated. The majority of the participants reported that the HCD was more challenging because of its larger volume, whereas the HFD was always easily ingested. The estimated caloric intake did not differ between the two diets, which was corroborated by the self-reported energy intake (Appendix A). One subject was able to only consume ~90% of each diet. The exact composition of the diets is provided in Appendix A. The average body weight was ~1.1% (0.8 kg) lower following the HFD compared to the HCD (Table 2). Furthermore, the HFD was associated with lower fasting triglyceride levels, whereas hemoglobin, LDL-, HDL- and total cholesterol were higher, compared to after the HCD (Table 2). All post-treatment values were within the normal ranges.

### 3.2. Appetite Scoring during the Mixed Meal Tests

On day 12, in each diet period, the participants ingested a standardized mixed meal test (MMT; consisting of 600 kcal brunch; 15E% protein, 31% carbohydrate, 54% fat; see Appendix A). At baseline, before the start of the MMT, the score for craving was increased after the HFD (Table 1). Following the MMT, scoring for hunger was higher at 30 min after the HFD compared to the HCD, which suggests that satiety was reached more rapidly after the HFD compared to the HCD. At 120 min after the start of the MMT, this difference was not significant anymore. No other differences were detected in appetite scoring during the MMTs (Table 1).

### 3.3. Glucose and Insulin Levels as Well as Incretins in Blood following the Mixed Meal Tests

Blood glucose and serum insulin concentrations were measured over 120 min after the MMT. Neither the AUC of glucose nor that of insulin levels differed between the HFD and HCD (Figure 1A,B). Furthermore, the HOMA-IRs calculated for baseline glucose and insulin values before the MMTs did not differ between the HFD and HCD (0.563 ± 0.10 vs. 0.508 ± 0.10, *p* = 0.083). All values measured were within normal reference ranges.

GLP-1 levels were almost doubled at 30 min (21.66 ± 2.24 vs. 13.28 ± 1.34, *p* = 0.0151) following the HFD compared to the HCD during the MTT (Figure 1C). However, the GIP levels did not differ during the MTTs after the two diets (Figure 1D). Thus, the HFD seemed to substantially and specifically increase GLP-1 levels without affecting insulin levels.

### 3.4. Other Systemic Metabolites Indicating Risk for a Diabetogenic Effect of High-Fat Diet

Analysis of the metabolomic profiles revealed that patterns of metabolites associated with glucose metabolism were differentially regulated during the HFD and HCD. Figure 2 illustrates the OPLS-DA (Orthogonal Projection to Latent Structures—Discriminant Analysis) loading that highlights the Bruker IVDr “good”-classified metabolites in this multi-dimensional model created to best describe the metabolomic separation between the HFD and the HCD.

The ketone bodies acetoacetate, acetone and β-hydroxybutyrate (β-HB) were increased after the HFD compared to the HCD. These changes likely reflect increased hepatic ketogenesis after the HFD after the fasting preceding the MMT (Figure 3A). After the ingestion of the MMT, the levels of all three ketone bodies decreased substantially, reflecting decreased hepatic ketogenesis upon refeeding after the overnight fast. However, all three ketones remained somewhat elevated after the HFD compared to the HCD throughout the MMTs (Figure 3A). The fasting values of acetone were negatively correlated with HOMA-IR (homeostatic model assessment—insulin resistance) during both the HFD and HCD (r-values −0.59, *p* = 0.023 for both; Table 3). Fasting β-HB was also negatively correlated with HOMA-IR during the HCD (r-value −0.53, *p* = 0.047; Table 3). The AUCs for acetoacetate and β-HB correlated negatively with insulin during the HFD (r-value −0.56, *p* = 0.036; Table 3) and with GLP-1 during the HCD (r-value −0.62, *p* = 0.016; Table 3), respectively.

Other metabolites that prominently differed between the diets were creatine and the branched-chain amino acids valine and leucine, which were all higher after the HFD compared to the HCD (Figure 3B). Three metabolites that have also been linked to insulin resistance, alanine, methionine and glutamine, were increased after the HCD compared to the HFD (Figure 3C). The AUC for alanine correlated positively with GLP-1 during the HFD (r-value 0.74, *p* = 0.002; Table 3), while the AUC for methionine correlated positively with GIP after the HCD (r-value 0.62, *p* = 0.016; Table 3).

The levels of the insulin resistance marker α-hydroxybutyrate (α-HB) are shown in Figure 4A. This metabolite was not included in the OPLS-DA loading model, based on the fact that it was not classified as “good” by the Bruker IVDr. The reason for this was that a majority of the samples (9/15) after the HCD did not show detectable levels of α-HB. In contrast, after the HFD, only one sample was below the detection limit. Thus, it seems that the HFD provoked the levels of α-HB to increase from undetectable to detectable levels in a majority of the healthy subjects. The AUC for α-HB did not show significant differences between the HFD and HCD, but the correlation analysis revealed that α-HB correlated negatively with insulin during the HFD (r-value −0.65, *p* = 0.011; Table 3). The serum levels of another intermediary metabolite, α-aminobutyrate (Figure 4B), were also excluded from the OPLS-DA loading model due to them not being classified as “good” based on only 6/15 of the baseline samples being above the detection limit of the assay after the HCD, whereas baseline levels after the HFD were measurable in 10/15 subjects. The AUC analysis revealed that α-aminobutyrate levels were higher (*p* = 0.022) after the HFD compared to the HCD (Figure 3B).

## 4. Discussion

The present study compared meal-induced glycemic control and metabolomic changes after a 2-week period of an HFD or HCD in random order in a cross-over study design in adult healthy, normal-weight humans. The diets were iso-caloric and fully prepared at the laboratory in order to achieve as well-controlled conditions as possible. They were combined with daily reports on the amount of food and liquids consumed. The differences between diets regarding body weight, hemoglobin, blood lipids and fasting insulin were small, and all values were within normal reference ranges. There were only minor differences in the preferability of the diets, with the subjects reporting the HFD to be easier to consume. After the HFD period, food craving was increased, and hunger decreased faster during the meal test.

There was no difference regarding glucose levels and insulin release between the HFD period and the equicaloric period with the HCD following an MMT. Actually, despite the increased GLP-1 levels after the HFD compared to the HCD, insulinemia was not changed. The levels of GIP were also not different between the two diets. It is intriguing that the meal-induced GLP-1 levels were so prominently increased after the HFD. The reason this did not result in increased insulin secretion is likely that glucose levels after the HFD were not significantly increased compared to the HCD, and therefore increased insulin levels were not necessary to keep glucose levels at bay as GLP-1 influences insulin levels in relation to glucose, e.g., during a glucose tolerance test [15]. The increased GLP-1 levels after the HFD likely contribute to the increased satiety during the MMT after the HFD compared to the HCD. One reason for the increased GLP-1 response after the HFD could be that the MMT used in the present study contained a high concentration of fat (see Appendix A), and that the enteroendocrine cells may have been primed to respond to fat by the HFD. Alternatively, the decreased GLP-1 effect on insulin secretion could be in line with an increased insulin resistance state during the HFD [16], as also suggested by the metabolomics data in the present study.

The marked increase in GLP-1 levels following the HFD found in the healthy controls should be placed in relation to our recently reported findings in obese subjects undergoing Roux-en-Y gastric bypass [17]. In that study, we suggested that the HFD, via increased intestinal ketone body production and inhibition of enteroendocrine cells in the jejunum, could inhibit GLP-1 and explain the decreased levels seen in obese subjects [17]. After Roux-en-Y gastric bypass, meal-induced GLP-1 levels increase prominently [18], and we suggested that this could be a result of a shutdown of jejunal ketone body production after the surgery [17]. This discrepancy between healthy, normal-weight subjects and obese patients remains to be examined in greater detail. The two studies differ in design, especially in temporal aspects. The GLP-1 response that we show in the present study may be lost with a longer duration of the HFD. It could also be that the physiological adaptation in the healthy individuals may be inherent and not the same as in individuals prone to develop obesity and diabetes in response to an HFD.

We measured the levels of the ketone bodies in the systemic venous blood, thus showing the total production of ketone bodies. We found that acetoacetate, acetone and β-hydroxybutyric acid were all markedly increased after the HFD compared to the HCD. This was true in the fasting state, but concentrations also remained higher throughout the MMT. This may reflect the insulin suppression and ketonemia during the overnight fasting preceding the test meals. The postprandially elevated ketone levels after the HFD vs. HCD could reflect an increased small intestinal ketogenesis induced by the HFD that we recently reported in obese patients before undergoing Roux-en-Y gastric bypass [17]. Even if the present level of evidence is low, the fact that the MMT after the HFD compared to the HCD increased the levels of ketone bodies and raised GLP-1 with maintained insulinemia in healthy persons is an observation of special interest. In obese subjects, the ketone bodies inhibit the GLP-1 response including insulinemia. This may speak in favor of a difference for the ketone body action on the GLP-1-producing cells in the intestines of obese and normal-weight volunteers.

The levels of branched-chain amino acids leucine and valine, as well as the metabolites creatine and α-HB, were increased by the HFD compared to the HCD. Elevated serum levels of branched-chain amino acids in epidemiological studies have been reported to predict the risk of insulin resistance and diabetes a long time prior to any other marker [5,6,7,9,10,19,20]. Higher plasma creatine has been shown to be associated with an increased risk of type 2 diabetes in males and with increased hepatic insulin resistance [21,22]. α-HB acid has been shown to be the top-ranked candidate to separate insulin-resistant from insulin-sensitive individuals in biochemical profiling studies in subjects without diabetes [23,24,25]. Circulating levels of α-HB were recently reported to be a robust marker of an elevated hepatic NADH/NAD^+^ ratio, which has previously been associated with impaired glucose tolerance and insulin resistance in humans [26,27]. Further, the improvement in insulin resistance after gastric bypass surgery has been shown to be correlated with a decline in α-HB [28].

The HCD resulted in higher serum levels of alanine, methionine and glutamine compared to the HFD. All three of these biomarkers have associations with the development of T2DM, but in different directions. Glutamine was found to be inversely correlated with the risk of T2DM [9,20]. Thus, the increased glutamine levels after the HCD are in line with other data in the present study to indicate that the HCD vs. HFD is less prone to promoting the risk for the development of insulin resistance. Alanine and methionine have been reported to be associated with insulin resistance and ultimately increased the risk of T2DM [9,29]. The AUCs for alanine after the HFD and for methionine after the HCD were positively correlated with the AUCs for incretins (GLP-1 and GIP, respectively), indicating these as biomarkers for changed insulin secretion. The AUCs for yet another metabolite, α-aminobutyric acid, were significantly higher after the HFD compared to the HCD during the MMTs. α-Aminobutyric acid is one of three isomers of aminobutyric acid that is also known as homoalanine, which is a metabolite of isoleucine biosynthesis. Isoleucine has been reported as a biomarker for the risk of T2DM and metabolic syndrome [20,30]. The exact relationship between isoleucine and α-aminobutyrate as biomarkers is unclear, but to our knowledge, α-aminobutyrate has not been previously reported to be correlated with insulin or incretin levels. When taken together, several of these intermediary metabolites indicated an oncoming insulin resistance following the HFD, even though there were no objective signs of a distorted glycemic control or insulinemia, compared to the HCD in healthy subjects.

The limitations of this study include the number of participants as well as the relatively short periods with the two diets. A significantly longer, e.g., 3–6 months, study period would be necessary in order to evaluate the actual long-term effects of these dietary macro-nutrient compositions. This would, however, come with the risk of compromising the strict adherence to the diet by the study subjects that we were able to achieve by the present study length and protocol.

Seen from the perspective that all study subjects were healthy and of a normal weight, the results are robust, and the power of our study is appropriate.

## 5. Conclusions

The present data show that despite well-maintained glycemia and insulinemia, meal-induced GLP-1 levels were higher after the HFD compared to the HCD. Furthermore, the metabolomic profiles (in particular, for branched-chain amino acids valine and leucine, and creatine, glutamine and α-HB) after the MMTs implied a metabolic shift towards impaired insulin sensitivity during the HFD compared to the HCD. It appears that the 2-week duration of the HFD placed the subjects at risk for diabetogenic development, but a significantly longer study period would be necessary in order to evaluate the actual long-term effects of these dietary macro-nutrient compositions.

## Figures and Tables

**Figure 1 nutrients-13-03322-f001:**
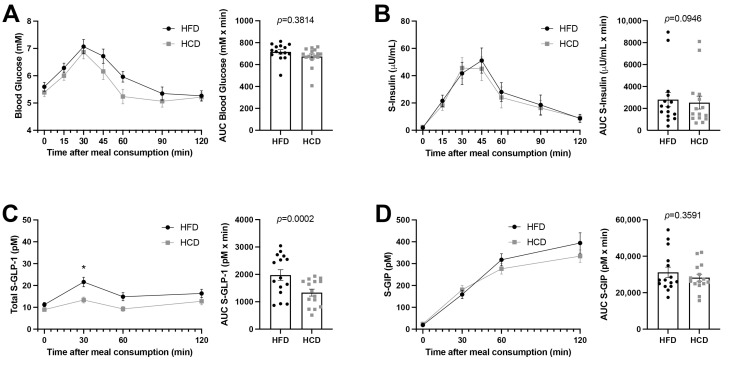
Glucose concentrations (**A**), insulin levels (**B**), GLP-1 (**C**) and GIP (**D**) in normal-weight, healthy subjects after two weeks of HFD (black circles) vs. HCD (gray squares) in a cross-over study design, 0–120 min after a mixed meal test (MMT). The MMT was ingested within the first 10 min starting at 0 min. * *p* < 0.05, using Sidak’s multiple comparisons test. The area under the curve (AUC), presented on the right in each panel, for each individual was calculated from the time profile and compared using the Wilcoxon test.

**Figure 2 nutrients-13-03322-f002:**
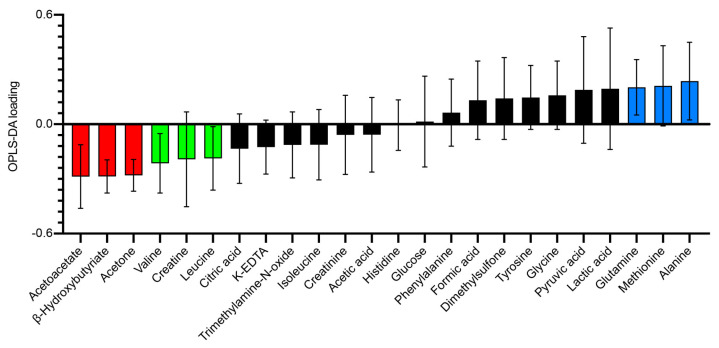
Loading plot of a one-component OPLS-DA model (R2X = 0.292, R2Y = 0.773, Q2 = 0.648) of baseline (HCD-HFD vs. HFD-HCD) metabolite values in normal-weight, healthy subjects on the day of the mixed meal test (MMT) after two weeks of HFD vs. HCD in a cross-over study design. Each individual was their own cross-validation subject. A total of 999 permutations returned R2 and Q2 intercepts at 0.181 and −0.279, respectively, meaning that the model is robust. Error bars indicate jack knife standard error of the loading computed from all rounds of cross-validation. The more a metabolite loading differs from 0, the more it influences the model. Corresponding data (HFD and HCD) cluster nicely in this PCA (principal component analysis) model without outliers in Hotelling’s T2 or DmodX (not shown). Metabolites classified as *good* (almost all individuals with a signal correlation value above 95%) from the Bruker IVDr analysis were used.

**Figure 3 nutrients-13-03322-f003:**
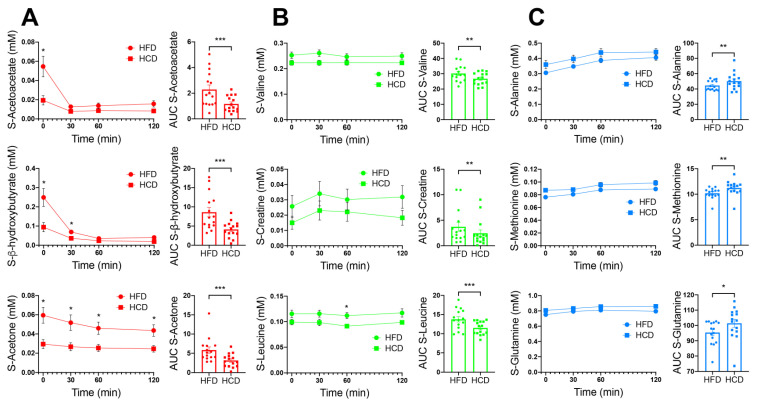
Serum levels of acetoacetate, β-hydroxybutyrate and acetone are shown in red (**A**), valine, creatine and leucine are shown in green (**B**) and alanine, methionine and glutamine are shown in blue (**C**); colors corresponding to the colors of metabolites in the OPLS-DA model in Figure 2), in normal-weight, healthy subjects after two weeks of HFD (circles) vs. HCD (squares), 0–120 min after an MMT. The MMT was ingested within the first 10 min starting at 0 min. * *p* < 0.05, using Sidak’s multiple comparison test. The area under the curve (AUC), presented on the right in each panel, for each individual was calculated from the time profile. * *p* < 0.05, ** *p* < 0.01, *** *p* < 0.001, using the Wilcoxon test.

**Figure 4 nutrients-13-03322-f004:**
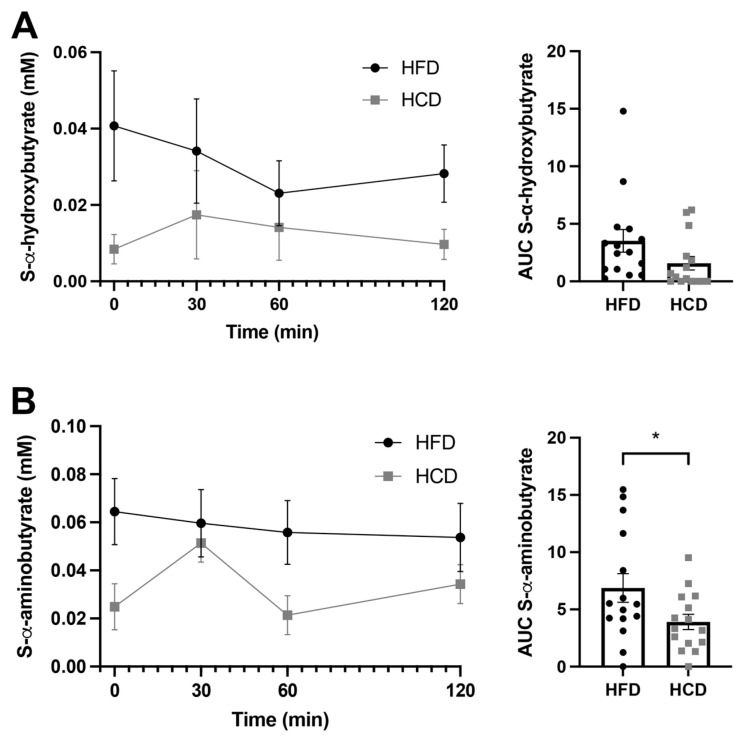
Serum levels of α-hydroxybutyrate (**A**) and α-aminobutyrate (**B**) in normal-weight, healthy subjects after two weeks of HFD (circles) vs. HCD (squares) in a cross-over design study, 0–120 min after a mixed meal test (MMT). The meal was ingested within the first 10 min starting at 0 min. The area under the curve (AUC), presented on the right in each panel, for each individual was calculated from the time profile. * *p* < 0.05, using the Wilcoxon test.

**Table 1 nutrients-13-03322-t001:** Appetite scoring before and after a standardized test meal.

	Scoring before Meal			Change from Baseline					
				30 min after Start of Meal			120 min after Start of Meal		
Sensation	HCD	HFD	*p*	HCD	HFD	*p*	HCD	HFD	*p*
Craving	48 (6.9)	67 (5.8)	0.021	21 (4.7)	19 (4.0)	0.44	39 (7.8)	22 (6.0)	0.48
Hunger	54 (6.6)	64 (6.4)	0.12	26 (5.7)	17 (3.8)	0.036	41 (7.3)	33 (6.0)	0.091
Satiation	16 (4.3)	13 (4.1)	0.37	60 (5.2)	68 (4.7)	0.14	45 (4.8)	56 (5.3)	0.092

**Table 2 nutrients-13-03322-t002:** Demographic and biochemistry profiles.

	Baseline Data	Post-Treatment Data		
		HCD	HFD	*p*
No. of randomized participants	16	15	15	
Gender (f/m)	8/8	7/8	7/8	
Age (years)	26.1 (0.8)			
Body weight (kg)	72.3 (3.0)	72.5 (3.0)	71.7 (2.9)	0.009
B-hemoglobin (g/L)	137 (2.9)	135 (2.9)	139 (2.7)	0.045
White cell count (10^9^/L)	5.0 (0.3)	5.3 (0.5)	4.8 (0.2)	ns
CRP (mg/L)	0.13 (0.09)	0 (0)	0.07 (0.07)	ns
Triacylglycerol (mmol/L)	0.93 (0.06)	0.96 (0.11)	0.70 (0.07)	<0.001
HDL (mmol/L)	1.72 (0.12)	1.47 (0.11)	1.82 (0.13)	<0.001
LDL (mmol/L)	2.94 (0.22)	2.42 (0.18)	2.79 (0.20)	0.003
LDL/HDL	1.75 (0.18)	1.77 (0.18)	1.65 (0.17)	ns
Total cholesterol	4.69 (0.88)	4.07 (0.72)	4.74 (0.88)	<0.001
ASAT (µkat/L)	0.44 (0.02)	0.41 (0.04)	0.45 (0.07)	ns
ALAT (µkat/L)	0.33 (0.02)	0.31 (0.03)	0.33 (0.03)	ns
ALP (µkat/L)	1.03 (0.07)	0.95 (0.07)	0.94 (0.07)	ns
Bilirubin (µmol/L)	10.6 (0.95)	10.9 (1.66)	10.2 (1.20)	ns
Na^+^(mmol/L)	140 (0.63)	139 (0.36)	140 (0.34)	ns
K^+^ (mmol/L)	4.2 (0.07)	4.1 (0.05)	4.0 (0.04)	ns
Creatinine (µmol/L)	80 (2.3)	84 (2.60)	85 (3.20)	ns

HCD: high-carbohydrate diet; HFD: high-fat diet; CRP: C-reactive protein (sensitive); HDL: high-density lipoprotein; LDL: low-density lipoprotein; ASAT: aspartate aminotransferase; ALAT: alanine aminotransferase; ALP: alkaline phosphatase; TAG: triacylglycerol. Mean (SEM), Wilcoxon’s test. All values were derived from fasting samples taken at baseline or on the day of the MMTs.

**Table 3 nutrients-13-03322-t003:** Correlations between metabolites and HOMA-IR, insulin and incretins.

Metabolite	Diet	Correlation with	r	*p*
Fasting value				
Acetone	HFD	HOMA-IR	−0.59	0.0227
Acetone	HCD	HOMA-IR	−0.59	0.0230
β-hydroxybutyric acid	HCD	HOMA-IR	−0.53	0.0471
AUC				
Acetoacetate	HFD	Insulin	−0.56	0.0335
Alanine	HFD	GLP-1	0.74	0.0022
α-hydroxybutyric acid	HFD	Insulin	−0.65	0.0107
β-hydroxybutyric acid	HCD	GLP-1	−0.62	0.0155
Methionine	HCD	GIP	0.62	0.0162

AUC = area under the curve. HOMA-IR = homeostatic model assessment for insulin resistance. Spearman non-parametric correlations. Fasting values and AUCs were derived from samples taken during MMTs.

## Data Availability

Datasets generated during and/or analyzed during the current study are not publicly available but are available from the corresponding author on reasonable request.

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
