# Peer review of "Glycemic Control and Metabolic Adaptation in Response to High-Fat versus High-Carbohydrate Diets—Data from a Randomized Cross-Over Study in Healthy Subjects"

_nutrients, 2021, doi:10.3390/nu13103322_

Round 1
Reviewer 1 Report
The topic is interesting, and I find pleasure to read the manuscript. The authors examined, glycemic control and metabolic adaptation in response to high fat versus high-carbohydrate diets. Authors used appropriate methodologies. The results are meaningful.
The findings are discussed in the context of relevant literature, and the strengths and limitations are well described.
However, I would like to suggest small changes.
Material and Method section- you can use sub heading to present your methodology clearly.
Please add your conclusion separately.
Author Response
Response to reviewer 1:
Dear Mr/Mrs
We would like to thank the reviewer for the kind review of our manuscript and the constructive suggestions which we have adhered to in the revised version of the manuscript as follows:
1) We have now added subheadings 2.1, 2.2... etc to the materials and methods section
2) We have added a separate conclusions chapter as heading no. 5.
With best regards
Ville Wallenius
MD, PhD, Associate Professor
Dept. Surgery, Inst. of Clinical Sciences, Sahlgrenska Academy,
University of Gothenburg, Gothenburg, Sweden.
Reviewer 2 Report
The ‘Abstract’ is concise and specific.
The ‘Introduction’ is appropriate to the topic.
In ‘Materials and methods' (or in supplementary materials) there should be a 1-day menu for each type of diet (HFD and HCD) provided. Information about the name of the Institute/ Hospital/ University where the studies were conducted should be also added. Kindly ask to add precisely the term of clinical studies (season, months, year).
The ‘Results’ were discussed adequately and sufficiently. At the end of the ‘Result’, it is worthwhile to propose the length of any future clinical studies to evaluate the actual long-term effects of these dietary macronutrient compositions.
Author Response
Response to reviewer 2:
Dear Mr/Mrs
We want to thank the reviewer for the constructive suggestions and we have adhered to these suggestions of reviewer 2 as follows:
1) We have now added an example of a daily menu for the high-carb and the high-fat diet as Supplementary Table 2.
2) As suggested we have added the setting (Institution and Hospital, season, months and year) for the study to the first paragraph of Materials and Methods, lines 64-66 (it was already stated under heading "Institutional Review Board Statement" but is now more visible to the reader).
3) We added a suggestion of length (3-6 months) for a future clinical study to evaluate the actual long-term effects of these dietary macronutrient compositions, but to the end of Discussion, lines 395-399. It felt that it may fit better to Discussion than Results.
With best regards
Ville Wallenius
MD, PhD, Associate Professor
Dept. Surgery, Inst. of Clinical Sciences, Sahlgrenska Academy,
University of Gothenburg, Gothenburg, Sweden.